# A Highly Efficient and Stable Photocatalyst; N-Doped ZnO/CNT Composite Thin Film Synthesized via Simple Sol-Gel Drop Coating Method

**DOI:** 10.3390/molecules26051470

**Published:** 2021-03-08

**Authors:** Md Elias, Md Nizam Uddin, Joyanta Kumar Saha, Md Awlad Hossain, Dali R. Sarker, Sonia Akter, Iqbal A. Siddiquey, Jamal Uddin

**Affiliations:** 1Department of Chemistry, Shahjalal University of Science and Technology, Sylhet 3114, Bangladesh; elias_chesust@yahoo.com (M.E.); daliranisarker.sust@gmail.com (D.R.S.); sonia_chesust@yahoo.com (S.A.); iqbal_siddiquey@yahoo.com (I.A.S.); 2Department of Chemistry, Jagannath University, Dhaka 1100, Bangladesh; joys643@yahoo.com (J.K.S.); awlad1975@yahoo.com (M.A.H.); 3Center for Nanotechnology, Department of Natural Sciences, Coppin State University, 2500 W. North Ave, Baltimore, MD 21216, USA; juddin@coppin.edu

**Keywords:** photocatalysis, thin film, DFT, photodegradation, lattice parameters, drop-coating

## Abstract

The thin film of N-doped ZnO/CNT nanocomposite was successfully fabricated on soda lime glass substrate by a simple sol-gel drop-coating method. The structural, morphological, chemical, and optical properties of as prepared samples were characterized by a variety of tools such as X-ray Diffraction (XRD), Field Emission Scanning Electron Microscopy (FE-SEM), Fourier Transform Infrared spectroscopy (FT-IR), and UV-visible spectroscopy. The hexagonal crystalline structure was confirmed from XRD measurement without any other impurity phase detection in samples. The N-doped ZnO/CNT composite showed excellent photo-catalytic activity towards cationic methylene blue (MB) dye degradation with 100% removal rate under UV light irradiation as compared to N-doped ZnO (65%) and pure ZnO (47.36%). The convincing performance has also been observed for the case of visible light irradiation. The enhancement of that photocatalytic activity might be due to narrowing the band gap as well as the reduction of electron–hole pair recombination in ZnO matrix with the incorporation of dopant nitrogen and CNT. It is assumed from the obtained results that N-doped ZnO/CNT nanocomposite thin film can be employed as an economically achievable and ecofriendly method to degrade dye with UV and visible light irradiation. Additionally, density functional theory (DFT) calculations were applied to explore the effect of N-doping on electronic structure of ZnO. The computational study has supported the experimental results of significant band gap contraction, which leads to the maximum absorption towards higher wavelength and no appreciable change of lattice parameters after doping. A conceivable photocatalytic mechanism of N-doped ZnO/CNT nanocomposite has been proposed as well.

## 1. Introduction

Organic pollutants like colored dyes, antibiotics, and colorless organic matters in aquatic environments are now considered as threats for the natural environment and human health [1,2,3]. Especially, colored dyes are mainly heterocyclic organic compounds, which have N=N bonds and sulfonic group in their molecular structure. These dyes are extensively used in the various industries such as the textile industry. The disposal of dyes into water bodies can constitute widespread damages to human health, aquatic life, and the food chain because of its noxious effects [4,5,6]. The dyes are highly colored toxic organic compounds, which can affect the photosynthesis action of entire ecosystem by reducing sunlight penetration and dissolve oxygen [7,8]. This indicates the necessity of treatment of water containing dye before disposal into the environment. A wide number of dye treatment methods such as coagulation [9], chemical oxidation [10], adsorption [11], reverse osmosis [12], ion exchange [13], ozonization [14], electrolysis [15], biodegradation [16], and photocatalytic degradation [17], etc. have been implemented to complete mineralization of dye.

Among these methods, heterogeneous photocatalysis by semiconductor photocatalysts is well known as the advanced oxidation process employed as a green and sustainable method for the detoxification of wide range organic dyes and inorganic pollutants into environmentally benign products at ambient temperatures and pressures [18]. This removal method is very simple as well as free from any involvement of toxic, hazardous, and expensive environmentally malignant chemicals [19].

Zinc Oxide (ZnO) has been regarded as a remarkable broad band gap semiconductor of 3.37 eV with substantial excitation binding energy (60 meV). In addition, ZnO based heterogeneous photocatalysis has been widely used for waste water treatment due to their outstanding photocatalytic property, abundance in nature, low cost, high chemical stability, appropriate optical band gap, low toxicity, and environmental sustainability [20,21,22]. Furthermore, some researchers reported that ZnO exhibited higher photocatalytic performance than TiO_2_ [23,24]. However, the ZnO photocatalyst still shows low photocatalytic activity because of several problems such as low photon utilization efficiency, fast recombination rate of charge carriers and narrow spectrum range due to wide band gap, etc. [25].

As a result, further modification of ZnO is indeed needed to improve the photocatalytic efficiency [26,27]. Doping with metals and non-metals into ZnO semiconductor has been used to modify the light absorption towards the visible region or to improve the photocatalytic efficiency. In particular, dopant nitrogen has been attracting much attention because of its abundance, low material toxicity, compatible size, low ionization energy, and ease of handling [28,29]. Zong et al. showed that N-doped ZnO bundle-like nanoparticles demonstrated a higher photocatalytic oxidation activity under visible light irradiation [28]. Qin et al. reported that heterostructure N-doped ZnO exhibited better photocatalytic degradation towards humic acids compared to pure ZnO thin film [30]. Moreover, many researchers have studied extensively on nitrogen doped various semiconductor photocatalysts such as TiO_2_, SnO_2_, CdS, etc. to improve their photocatalytic activity [11,29,31,32,33,34]. It can reduce the band gap and also decrease the rate of electron–hole pair recombination.

Another approach of improving the photocatalytic efficiency of ZnO semiconductor photocatalyst involves the attachment of carbon nanotubes(CNTs) [35]. CNTs have some remarkable advantages, properties like superior chemical stability, strong adsorptive ability, and excellent electronic and conductive properties [26]. Additionally, the large electron storage capacity of CNTs improves the photocatalytic efficiency by accepting the photogenerated electron from the conduction band of ZnO at the heterojunction interface of ZnO and CNT [36]. From the several studies, it is concluded that polar groups on the surface of CNT due to functionalization provide a significant way to enhance the photocatalytic efficiency of the nanocomposite thin film photocatalysts [36,37]. Zhu et al. prepared ZnO–coated multi-walled CNT composites via a sol process and observed improved photocatalytic degradation of methyl orange (MO) dye (~98%) than pure ZnO [38]. Azqhandi et al. fabricated Cd-doped ZnO/CNT nanocomposites by a microwave assisted hydrothermal method and investigated their photocatalytic behavior under UV light irradiation and adsorption properties [39].

Despite the above significant advancement, numerous issues related to the powder form photocatalysts and fabrication techniques remain to be addressed. For example, the filtration process is necessary before reuse and recycles, although thin film forms of those materials is the key to minimizing the issue. Moreover, many thin film fabrication methods such as dip coating, chemical vapor deposition, spray-pyrolysis, electro-deposition, chemical bath deposition, and other methods have been used in the last few decades [40]. The newly proposed drop coating thin film fabrication technique is very easy, low cost, as well as the amount of materials for each sample could be controlled at a molecular level. Very recently, we have reported the sol-gel drop coating synthesis of Ce-doped ZnO/CNT composite thin film and found the enhanced visible light photodegradation over MB dye. Despite there being a huge number of works on composite photocatalysts, no work has been reported on N-doped ZnO/CNT composite thin film.

In this research, the N-doped ZnO/CNT thin film on glass substrate has been fabricated by a simple sol-gel drop coating method. The influence of dopant nitrogen and sensitizer CNTs on the structural behavior of the ZnO was analyzed by XRD, FESEM, FT-IR, DFT, and UV-visible spectroscopy. The prepared nanocomposite thin film demonstrated improved photocatalytic efficiency for the degradation of MB dye under UV and visible light irradiation as compared to pure ZnO and N-doped ZnO thin films.

## 2. Results and Discussion

### 2.1. XRD Analysis

The XRD patterns were used to determine the effects of the addition of dopant on the crystal phase, lattice strain, lattice dislocation density, and crystallinity of ZnO thin films. Figure 1 shows the XRD patterns of (a) pure ZnO, (b) N-doped ZnO, and (c) N-doped ZnO/CNT thin films, respectively. All the prepared thin films with high crystallinity revealed a hexagonal wurtzite crystal structure, which mostly matches well with the reported data (JCPDS No. 01-075-0576) [35]. The peaks centered at an angle (2θ) of 31.91°, 34.55°, 36.29°, 47.75°, 56.89°, 63.13°, 66.47°, 68.07°, and 69.33° corresponding to (100), (002), (101), (102), (110), (103), (200), (112), and (201) planes, respectively. The peaks at 25.7° and 42.5° correspond to the (002) and (100) planes of CNT (Figure 1c). The diffraction peaks of the pure ZnO are sharp and intense, revealing the highly crystalline character of the ZnO sample, while the diffraction peaks of the N-doped ZnO and N-doped ZnO/CNT are broad and weak (Figure 1b,c), representing a small crystal size of these thin films. The most intense peak (36.29°) of the XRD pattern of bare ZnO thin films has been little shifted to higher 2θ values for the N-doped and CNT composite of that materials. This confirms the doping of ZnO with nitrogen and composite with CNT. In addition, the XRD data were employed to evaluate average size of the crystallites, D from the peak half-width *β*, using the Scherrer’s equation [21] (Equation (1)):(1)$$D=\frac{k\lambda }{\beta \cos\theta }$$
where *k*, *λ,* and *θ* are the shape factor of the particle, wavelength, and the incident angle of the X-rays, respectively. According to this Scherrer equation, the average crystallite size for ZnO, N-doped ZnO, and N-doped ZnO/CNTs is 29.07 nm, 27.13 nm, and 14.28 nm, respectively. The crystalline sizes have been reduced significantly after addition of dopant nitrogen as well as CNT compared to that of pure ZnO. These results indicate that only doping of nitrogen into ZnO matrix plays a role to reduce the particle size in N-doped ZnO. Moreover, CNT inhibits the growth of ZnO crystallites drastically in N-doped ZnO/CNT nanocomposite thin film [8].

The lattice strain induced due to the crystal imperfection and distortion in the pure ZnO thin film, N-doped ZnO, and N-doped ZnO/CNT was calculated by the Stokes–Wilson equation (Equation (2)):Ε = β/4tanθ(2)

The dislocation density points towards the amount of crystallographic defect or irregularity present in the crystal that strongly influences the properties of the synthesized materials. The dislocation density was calculated by applying the Williamson–Smallman relation [41] (Equation (3)):δ = 1/D^2^(3)

The average crystallite size, lattice strain, and dislocation density results are summarized in Table 1.

The lattice constants a & c are related to the inter-planar spacing of the atomic planes d for hexagonal wurzite structure and can be calculated from the XRD spectra by the following equation [41] (Equation (4)):(4)$$\frac{1}{d_{hkl}^{2}}=\frac{4}{3}\left(\frac{h^{2}+hk+k^{2}}{a^{2}}\right)+\frac{l^{2}}{c^{2}}\;$$

The Miller indices h, k, & l of (100) plane of hexagonal structure is used to calculate the lattice constant *a* with the help of the following equation [40] (Equation (5)):(5)$$a=\frac{\lambda }{\surd 3\sin\theta }\;$$

The lattice constant c is calculated from the reflection (002) plane by using the following equation [42] (Equation (6)):(6)$$c=\frac{\lambda }{\sin\theta }\;\;\;\;$$

Finally, the unit cell volume of hexagonal pure ZnO, N-doped ZnO, and N-doped ZnO/CNT thin films is calculated by the following equation [43] (Equation (7)):(7)$$V=\frac{\surd 3}{2}a^{2}c\;\;\;$$
where *a* & *c* are the lattice constants of the prepared thin films and the obtained results of the lattice constants and the unit cell volume of synthesized thin films are summarized in Table 2. The lattice parameters value is the indication of positive (extensive) or negative (compressive) stress of crystallites if the value is lower or higher than the bulk materials [40]. The value of *c* is less than the value of bulk ZnO implies that all the thin films of ZnO crystallites are the state of compressive stress. The lattice constants a = b and c of hexagonal unit cell exhibit lower value compared with the value of bulk ZnO. It can be seen from Table 2 that the lower unit cell volume of all the prepared thin films compared to the standard is a result of higher stress.

### 2.2. FESEM Analysis

Figure 2 shows the FESEM images of as synthesized (a) pure ZnO, (b) N-doped ZnO, and N-doped ZnO/CNT thin films calcined at 500 °C. The surface morphology of pure ZnO thin film reveals the irregular spherical shape of particles with some agglomeration while N-doped ZnO shows a smooth surface with small spherical grains distributed as uniform, compact, and adherent to substrate. However, the morphology of N-doped ZnO/CNT is formed in a more consistent spherical shape with a lower degree of agglomeration. The comparatively homogeneous morphology (Figure 2c) might be due to the inclusion of functionalized CNT that contains the polar functional groups on its surface that impede the agglomeration of nanoparticles. However, it looks very difficult to figure out the CNT structure from Figure 3c with that low resolution image. In addition, the uniform and small size of the catalyst particle can contribute to a greater adsorption of dye on surface and more light absorption in this photocatalytic system. With the inclusion of dopant nitrogen and CNT, the grain size decreases from 30 to 14 nm. The unevenness and grain size decrease because of the doping nitrogen as well as CNT that ascribe the enhancement in crystalline quality.

### 2.3. FT-IR Analysis

The FT-IR spectrum illustrates the information of a particular compound about its functional groups, molecular geometry, and inter/intramolecular interactions. Figure 3 displays the FT-IR spectra of as prepared (a) pure ZnO, (b) N-doped ZnO, and (c) N-doped ZnO/CNT nanocomposite thin film, respectively, and those were performed in the wave number range 400–4000 cm^−1^ using the KBr method at room temperature. All FT-IR spectra display a broad absorption band at around 3400–3600 cm^−1^ due to the presence of the O–H stretching mode of the hydroxyl groups. The absorption band around 2800–2900 cm^−1^ is identified as a stretching mode for the C-H bond of binder TEA. The band at 1650 cm^−1^ characterizes the H–O–H bending vibration of water. The bands in the low wave number region, 426 cm^−1^, correspond to the vibration modes of Zn–O [41]. It is assumed that the bands appearing at 1109 cm^−1^ and 1020 cm^−1^ are due to stretching vibrations of Zn-O-Si and Si-O-Si that appear from the glass substrate [44].

### 2.4. UV-Visible Absorption Spectra Analysis

The photocatalytic activity of photocatalyst depends on the light absorption capacity. UV-visible absorption spectra have been measured to determine the optical properties of the prepared thin films and the obtained results are shown in Figure 4a. The pure ZnO has an absorption maximum at 362 nm, whereas the peaks at 368 nm and 366 nm are obtained for the N-doped ZnO/CNT and N-doped ZnO, respectively. The presence of absorption peak indicates the electronic transition from valence band to conduction band of ZnO nanoparticles [45]. It can be seen that N-doping has an influence on the absorption spectrum into UV to the visible region (red shift). The similar results are also found in the references for the redshift in the absorption edge [30]. The band gap of prepared thin films has been calculated with help of the following Tauc equation (Equation (8)):αhν= (hν − Eg)1/2(8)
where Eg is the band gap energy, hν is the photon energy, and α refers to the absorption coefficient, by plotting the value αhν along the *y*-axis and hν along the *x*-axis followed by the extrapolating the liner portion of the curve to hν axis (αhν)^2^ = 0 (Figure 4b). The estimated band gap of pure ZnO, N-doped ZnO and N-doped ZnO/CNT is 3.18 eV, 2.90 eV and 2.65 eV, respectively.

### 2.5. Density Functional Theory (DFT) Study

The lattice parameters of optimized Zn_8_O_8_ a = b = 3.127Å and c = 5.032 Å are in good agreement with experimental values (Table 2). After nitrogen doping, the lattice parameters of N-doped Zn_8_O_8_ (a = b = 3.124 Å and c = 5.098 Å) changes 0.003 Å and 0.066 Å for a and c, respectively. Theoretically calculated lattice parameters deviate 3.3% from those that are experimentally measured. To determine the effect of nitrogen doping in ZnO, band structure and total density of states (TDOS) of ZnO and N-doped ZnO have been calculated as shown in Figure 5. The band structure shows the band gap of ZnO is 2.18 eV, which is lower than that experimentally obtained. This deviation is originated because the DFT calculation is concerned with only the ground state [46]. After nitrogen doping, the band gap reduces to 1.71 eV, which is relevant for UV-visible absorption spectrum. To justify this, we calculated the total density of states (TDOS). The peak of the valence band maximum (VBM) of ZnO is located below the Fermi level. However, nitrogen doping in ZnO attributes the upward shift of VBM to reduce the band gap. There is almost no effect of nitrogen doping on conduction band minimum (CBM) of ZnO.

### 2.6. Photocatalytic Activity Evaluation

The photocatalytic performance of pure ZnO, N-doped ZnO, and N-doped ZnO/CNT thin films in degradation of MB dye was carried out under UV and visible light irradiation and results are shown in Figure 6. To confirm the photolysis and adsorption onto thin film, the adsorption as well as photodegradation experiments were conducted separately under visible light and UV light irradiation for 4 h. For both studies, no appreciable changes were observed due to adsorption and self photodegradation of the MB dye. About 100% of MB is degraded by an N-doped ZnO/CNT nanocomposite thin film photocatalyst within 4 h under UV light irradiation, while the efficiency is 63.69% under visible light irradiation. From Figure 6, it was noticed that about 60% of MB was removed after 60 min by UV light irradiation for N-doped ZnO/CNT, whereas only 16.66% and 28.57% for pure ZnO and N-doped ZnO photocatalysts. In case of visible light irradiation, the effectiveness of pure ZnO has 23.41% degradation efficiency, much less compared with N-doped ZnO (34.62%) and N-doped ZnO/CNT (63.69%). These results clarify that N doping enhances the photocatalytic efficiency of plain ZnO by reducing the band gap. On the other hand, CNTs improved the interfacial electron transfer rate and lessened the recombination possibility between photo-induced electron hole pair (e^−^/h^+^) more effectively than the ZnO photocatalyst alone. It could be summarized that the reduction of e^−^/h^+^ recombination and generation of more **^•^**OH radicals in the N doped ZnO as well as N-doped ZnO/CNT samples play a vital role in the improved rate of photocatalytic degradation. The produced **^•^**OH radicals have strong enough reactivity to break different (C-C, C=C, and C=O) bonds in MB molecules lead to the formation of CO_2_ and H_2_O at the end. However, many researchers have used numerous pollutants such as 4-cholorophenol, antibiotics, and natural organic matters, in order to examine the feasibility of the photocatalyst [2,47,48].

Recent reports point out that several oxidants such as peroxymonosulfate, hydrogen peroxide (H_2_O_2_), peroxydisulfate, ClO_3_^−^ etc. in heterogeneous photocatalytic system increase the degradation efficiency either by lowering e^−^/h^+^ recombination rate through accepting the conduction band electrons at the surface of ZnO photocatalyst or by providing additional oxygen atoms as an electron acceptor to form the superoxide radical ion (O_2_^•−^) [48]. Herein, the effect of varying dose of oxidant H_2_O_2_ (5–30 mmol) was studied in case of N-doped ZnO/CNT nanocomposite photocatalyst thin film for MB dye degradation under UV light irradiation. The results showed that MB dye degradation increases due to the addition of H_2_O_2_ and the MB dye degradation efficiency was 100% within 60 min under UV light irradiation for 5 mmol H_2_O_2_. The results of degradation efficiency were similar up to 25 mmol H_2_O_2_, whereas, for additional doses like above 25 mmol of H_2_O_2_, the degradation efficiency was not changed. The higher degree of degradation is due to the formation of highly reactive radical intermediates and the electron capture by oxidant H_2_O_2_. As a consequence, the addition of H_2_O_2_ to the heterogeneous system increases the concentration of ^•^OH radicals.

N-doped ZnO/CNT (electron from conduction band) + H_2_O_2_ → N-dopedZnO/CNT + OH^–^ + OH^•^

As H_2_O_2_ is an electron acceptor species, it does not only produce ^•^OH radicals but also obstructs the electron hole recombination [48]. When the H_2_O_2_ concentration becomes high, the excess H_2_O_2_ consumes hydroxyl radicals as scavenger [49] (Equation (9)):H_2_O_2_ + ^•^OH → HO_2_^•^ + H_2_O(9)

When H_2_O_2_ was added, the maximum degradation was achieved in the first 60 min only, whereas, without oxidants, it took 240 min for the 100% degradation. The influence of initial dye concentration on photocatalytic degradation is defined by the Langmuir–Hinshelwood kinetic model (Equation (10)) [50]:(10)$$r\;=-\frac{\left[MB\right]}{dt}\;=\;-\frac{kK\left[MB\right]}{1+K\left[MB\right]}$$
where r is the rate of reaction, *k* is the reaction rate constant (mgL^−1^ min^−1^), *K* is the observed equilibrium constant of the reactant (Lmg^−1^), and [*MB*] is the reactant concentration (mg L^−1^). When [*MB*] is very small, *K*[*MB*] is negligible with respect to unity and photocatalysis can be simplified to an apparent pseudo-first-order kinetics [51]. Therefore,
(11)$$r\;=-\frac{\left[MB\right]}{dt}\;=\;kK[MB]$$

Integration of Equation (11) gives
(12)$$\ln\left(\frac{{\left[MB\right]}_{0}}{\left[MB\right]}\right)=k_{obs}t$$
where *k*_obs_ is the observed pseudo-first-order rate constant (min^−1^). A plot of $\ln\left(\frac{{\left[MB\right]}_{0}}{\left[MB\right]}\right)$ vs. irradiation time *t* provided a straight line with their rate constants at different initial concentrations of *MB* (Figure 7). The observed degradation rate and correlation coefficients are represented in Table 3. All correlation coefficients values suggest that the proposed kinetic model was in excellent agreement with our experimental data. N-doped ZnO/CNT under UV light irradiation shows the best photocatalytic performance with a rate constant much higher than those of N-doped ZnO and bare ZnO under visible as well as UV light irradiation. The same result was observed by Hadi et al. during the photo-catalytic degradation of MB dyes using Tungsten doped zinc oxide as photocatalysts [52].

### 2.7. Study of Photocatalyst Stability and Reusability

The long-term stability and recyclability of photocatalyst thin film are very crucial for MB dye degradation as well as large scale practical utilization in the industry. Therefore, the reusability of a prepared N-doped ZnO/CNT nanocomposite thin film was examined by five consecutive cycles for the degradation of MB dye under UV light illumination, and results are shown in Figure 8. After every test, the thin film composite photocatalyst was washed with distilled water and dried naturally at room temperature followed by the heat treatment at 500 °C for 1 h. After that, a thin film was applied for a second cycle MB dye degradation test like the first cycle experiment. The degradation efficiency of MB dye was reduced to about 5% after five successive cycles, which means that degradation efficiency reached 95% from 100% after five recycling times. These results definitely indicate that the fabricated N-doped ZnO/CNT nanocomposite thin film photocatalyst is highly stable and reusable. Furthermore, a comparison of the photocatalytic activity of the N-doped ZnO/CNT thin film photocatalyst with previously reported photocatalysts was listed in Table 4. From the Table 4, it is clear that degradation efficiency and reusability of N-doped ZnO/CNT nanocomposites are comparable to the results with earlier reported ZnO-based materials.

### 2.8. Photocatalytic Degradation Mechanism

The excellent photocatalytic efficiency of synthesized N-doped ZnO/CNT composite supports the following degradation mechanism of MB dye under UV light irradiation as described by Chemical equations (1–7). The computational and theoretical studies confirm the incorporation of dopant N into the ZnO matrix and doesn’t change the wurzite structure of ZnO; it only replaces the O atom with an N atom. Upon illumination of N-doped ZnO/CNT composite semiconductor photocatalysts by the UV light having energy (*hv*>>E_g_) greater than the band gap results in the formation of charge carriers’ electron-holes (e^−^/h_+_). These electron–holes (e^−^/h_+_) recombine quickly, thereby producing minute amounts of highly reactive oxidizing species such as hydroxyl radicals, hydrogen peroxide, and superoxides that take part in the photocatalytic degradation of MB dye. However, the addition of CNT with the N-doped ZnO forms a heterojunction interface that can reduce the electron-hole recombination rate:(1)N-doped ZnO/CNT composite + *hv*→ h_vb_^+^ + e_cb_^−^(CNT)(2)H_2_O + h^+^ → H^+^ + OH^•^(3)OH^−^ + h^+^ → OH^•^(4)O_2_ + e^−^(CNT) → O_2_^•−^(5)O_2_^•−^ + H_2_O + H^+^ → H_2_O_2_ + OH^−^(6)H_2_O_2_ + e^−^ → OH^•^+ OH^−^
(7)OH^•^+ MB→ H_2_O + CO_2_


The improved degradation efficiency of N-doped ZnO/CNT composite is due to the synergistic effect of CNT into N-doped ZnO as shown in Figure 9.

## 3. Experimental Section

### 3.1. Materials

Multi-walled carbon nanotubes (MWCNT) were collected from Sigma-Aldrich (St. Loius, MO, USA) with the particulars as follows: length (5–9 μm), diameter (110–170 nm) and assay >90% (carbon basis). Zinc acetate dihydrate (Zn(CH_3_COO)_2_.2H_2_O; Merck, Darmstadt, Germany), tri-ethylamine (TEA) (≥99%, Sigma Aldrich, St. Loius, MO, USA) were purchased to use as Zn provider and stabilizing agent, respectively. Urea, hydrogen peroxide, sulfuric acid, nitric acid, and methylene blue (MB) dye were acquired from Sigma Aldrich (St. Loius, MO, USA). All other reagents were analytical grade and used as received.

### 3.2. Functionalization of CNT

Functionalization of CNT leads to the oxygenated polar functional groups on its surface. Therefore, oxygenated polar functionalities enhance the binding of CNT with ZnO through some physical/chemical interactions, for instance hydrogen bonding, van der Waals attraction as well as other bonds. In addition, the acid treatments (functionalization) also reduce the impurities of the amorphous carbon as well as unfold the ends and or break the tubes. In a typical functionalization, fresh MWCNT (1 g) was immersed into 40 mL of mixed solution (1:1 ratio) of concentrated sulfuric acid and nitric acid. [35]. Then, the combined solution was refluxed for 5 h at 60–70 °C in an oil bath to get dark-brown suspension. After completion of reaction, the prepared suspension was separated by centrifugation after cooling at room temperature. Thereafter, the product was washed with de-ionized water and ethanol until the pH of the filtrate become neutral, and dried in vacuum oven at 80 °C to use in the next step.

### 3.3. Fabrication of N-Doped ZnO/CNT Thin Film

N-doped ZnO/CNT thin film was fabricated by applying sol-gel drop coating protocol according to the literature with some modifications [57]. Typically, to prepare the N-doped ZnO/CNT, 0.7316 g of zinc acetate dihydrate and 0.4 mL of tri-ethylamine were first dissolved in 9.6 mL of anhydrous ethanol under stirring environment (300 rpm) for 4–5 min, followed by the addition of 5wt% of nitrogen (urea) (28 mg urea) and 5wt% of CNTs (5 mg) to the above solution. The resultant solution was sonicated (Powersonic 405, Hwashin Technology Co., Seoul, Korea) for 1 h to obtain a uniform dispersion. Then, the solution was preserved for 1 h maintaining temperature (55 ± 5) °C with continuous stirring at 300 rpm until a quite stable and transparent sol was obtained. Prior to the coating process, the stable N-doped ZnO/CNT sol was aged for 24 h and a commercial bench grinder (model: ST-150) was used to abrade glass slide having dimension of 10 mm × 60 mm × 1.5 mm. The abrasive glass slide was cleaned by using potassium dichromate and dichloromethane solution, respectively, followed by washing with alcohol and distilled water and then dried at 100 °C in an oven for further use. The N-doped ZnO/CNT nanocomposite thin film (heterostructure) was fabricated by a sol-gel drop coating technique, as follows: first, the 0.4 mL sol was used drop-wise on a rubbed glass substrate using a pipette. Subsequently, the sol supported on substrate was warmed for 20 min in an oven at 80 °C to eliminate the loosely bonded particles from the glass surface. Coated thin film could cover approximately 2 µm on the above estimated area of abrasive glass substrate. To obtain the thick film (~4 µm), the coating was repeated twice as well as calcined using a muffle furnace (JSMF-30T, Gongju, Korea) retaining temperature 500 °C for 2 h. Similarly, the N-doped ZnO thin film was prepared in the absence of CNT. Finally, pure ZnO thin film was prepared using the same procedure. A schematic flow diagram for the fabrication of N-doped ZnO/CNT nanocomposite thin film by sol-gel drop coating method is shown in Figure 10.

### 3.4. Film Characterization

The crystallinity of the prepared thin film was investigated using the XPERT PRO X-ray diffraction (XRD, Almelo, Netherlands) instrument with Cu-K_α_ monochromatic radiation source (λ = 1.5406 Å, V = 40 kV, I = 30 mA, RT) in the 2θ limit of 10–70°. The average crystallite sizes of synthesized thin films were estimated with the help of Scherrer equation. The surface morphology and microstructure of synthesized nanocomposite thin films were assessed by the Field Emission Scanning Electron Microscope (FESEM; JSM-7600F, Tokyo, Japan). The optical properties of samples were accomplished by double beam UV-Visible spectrophotometer (UV-1800, Shimadzu, Kyoto, Japan). The Fourier Transform Infrared (FT-IR) spectra of the prepared thin films were recorded in transmission mode on KBr pellets using a spectrometer (Shimadzu IR Prestige 21, Kyoto, Japan).

### 3.5. Photo-Catalytic Assessment

The photo-catalytic degradation efficiency of as-fabricated nanocomposite thin film was determined by measuring the photodegradation of an aqueous solution of MB dye as a target pollutant under visible light and UV light irradiation. The experimental setup is described as follows: firstly, to evaluate photo-catalytic activity of the prepared thin films under visible light irradiation, 200 mL aqueous solution of MB (5 mg/L) and a thin film photocatalyst were taken in a beaker. Then, the beaker containing catalyst was placed in a reactor consisting with a cycled cooling water system to avoid any thermal reaction. In each study, the aqueous solution of MB containing catalyst was initially stirred vigorously for 60 min in dark to attain adsorption-desorption equilibrium before exposure to visible light (200W tungsten lamp). At given time intervals, the solution was taken from the reaction container with a pipette and the change of residual concentration of MB dye solution was monitored by UV-visible spectrophotometer (UV-1800, Shimadzu, Kyoto, Japan) at a characteristics wavelength of 664 nm. The photocatalytic activity of the fabricated thin films over MB dye was analyzed with the help of the following Equation (13):Degradation Efficiency (%) = (C_0_ − C_t_)/C_0_ × 100(13)
where C_0_ is the initial concentration of MB before light irradiation and C_t_ is the concentration after light irradiation at time t. Similarly, the photodegradation efficiency of thin films photocatalysts under UV light irradiation was carried out by the above-mentioned method using the 60 W low pressure mercury lamp. The effect of oxidant H_2_O_2_ addition on photodegradation efficiency over MB dye under UV light illumination and thin films reusability were also investigated.

### 3.6. Computational Methods

To investigate the nitrogen doping effect on ZnO, we studied the electronic properties of ZnO and N-doped ZnO. Consequently, a hexagonal wurtzite structure of ZnO with the space group P6_3_mc and $C_{6v}^{4}$ symmetry was modeled. The size of supercell 2 × 2 × 1 was considered for 16-atom Zn_8_O_8_. We replaced one oxygen atom by a nitrogen to get 6.25% N-doped ZnO (Zn_8_N_1_O_7_). The cell volume and atomic relaxation for both Zn_8_O_8_ and Zn_8_N_1_O_7_ were carried out until the forces on each atom were below 0.01 eVÅ^°−1^. The total energy of the system was converged within 10^−5^ eV. The plane waves were expanded up to a cutoff energy of 450 eV. The Brillouin zone was considered using 4×4×1 k-point Monkhorst–Pack mesh. The exchange–correlation interaction in the generalized gradient approximation (GGA)+U method with Perdew, Burke, and Ernzerh of (PBE) function was used for geometry optimization and energy calculation [58]. The effective Hubbard U values 10 and 6 were used for Zn-3d and O-2p, respectively [59]. We adopted the same U value of O for N atom in Zn_8_N_1_O_7_. The valence electron configurations Zn3d^10^4s^2^, O2s^2^2p^4^ and N2s^2^2p^3^ were considered for pseudo potential construction. All density functional theory (DFT) based calculations were performed using the projector augmented wave (PAW) pseudo potentials, as implemented in the VASP code [60].

## 4. Conclusions

A thin film of N-doped ZnO/CNT nanocomposite was fabricated by a low cost sol-gel drop coating protocol and used to achieve photo-catalytic degradation of MB dye under visible and UV light irradiation, respectively. Experimental results of the XRD and FESEM studies indicated that ZnO was formed in nano scale successfully without any impurities in the hexagonal wurzite crystal phase. Findings of experimental and computational studies of lattice parameters of prepared thin films with the addition of nitrogen and CNT into the ZnO crystal do not show any significant differences, which indicate no change of crystal lattice of ZnO. Nearly 100% degradation of a 5 mgL^−1^ MB dye solution at neutral pH was achieved by an N-doped ZnO/CNT photocatalyst after 4 h of irradiation by UV light. The H_2_O_2_ oxidant effect revealed the 100% MB dye degradation efficiency for an N-doped ZnO/CNT nanocomposite thin film within 60 min under UV light irradiation. The N-doped ZnO/CNT thin film may possibly be reused, which means that the photo-catalytic degradation process could be functioned at a fairly low cost. Overall, the findings in this report can be beneficial and supportive in designing up a scalable and practical way for industrial wastewater treatment.

## Figures and Tables

**Figure 1 molecules-26-01470-f001:**
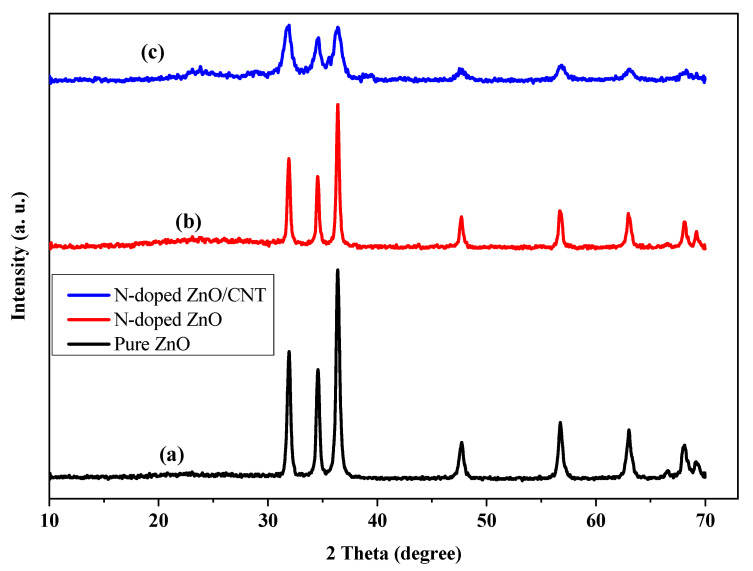
XRD patterns of (**a**) pure ZnO, (**b**) N-doped ZnO, and (**c**) N-doped ZnO/CNT thin film.

**Figure 2 molecules-26-01470-f002:**
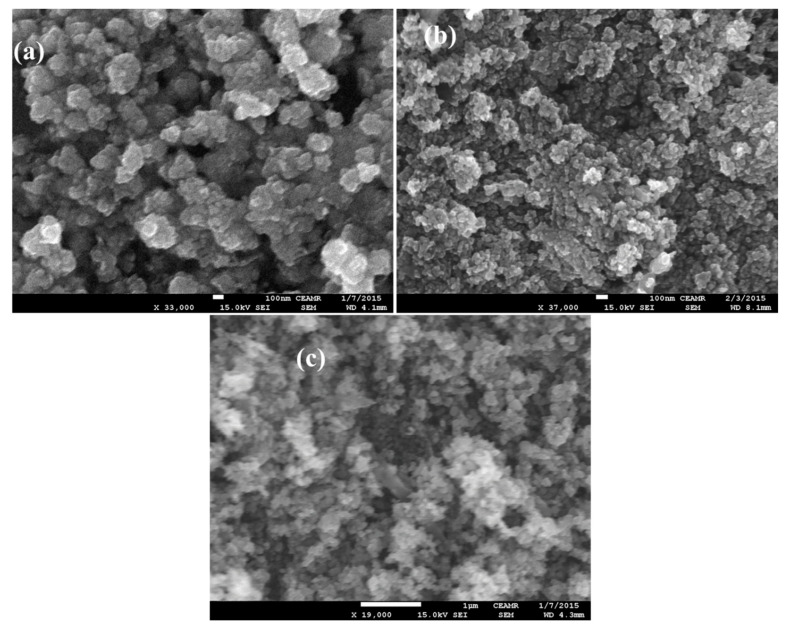
FESEM images of (**a**) pure ZnO; (**b**) N-doped ZnO; and (**c**) N-doped ZnO/CNT nanocomposite thin films.

**Figure 3 molecules-26-01470-f003:**
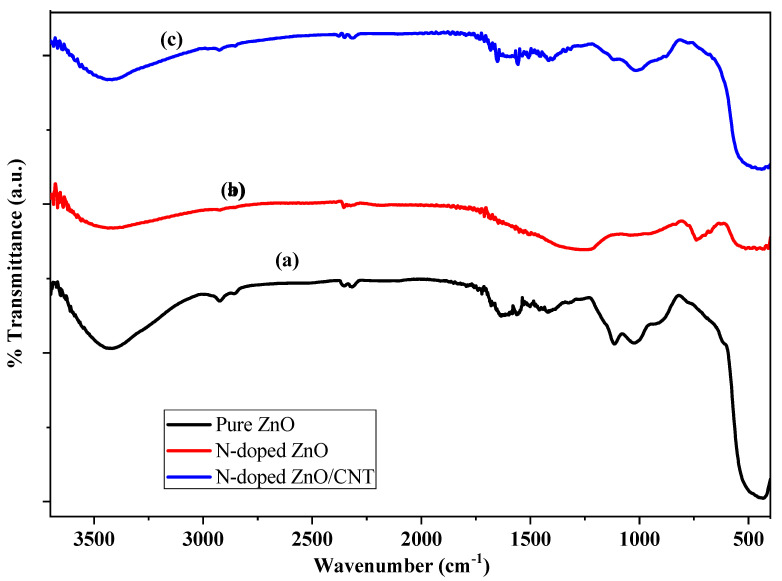
FT-IR spectra of (**a**) pure ZnO; (**b**) N-doped ZnO; and (**c**) N-doped ZnO/CNT nanocomposite thin films.

**Figure 4 molecules-26-01470-f004:**
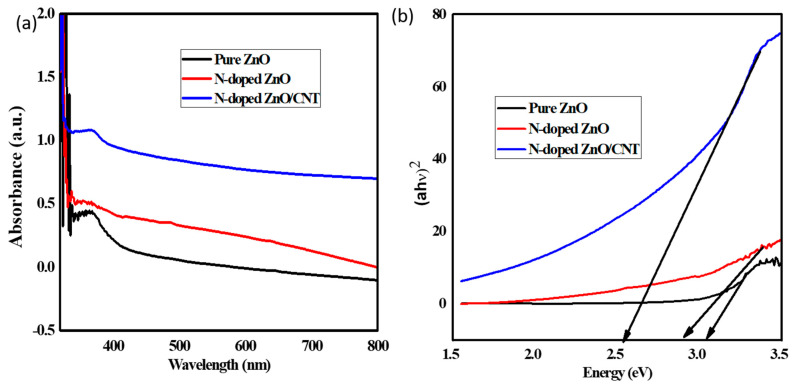
(**a**) UV-vis absorption spectra of pure ZnO, N-doped ZnO and N-doped ZnO/CNT thin films; (**b**) calculated band gap from Tauc plot of pure ZnO, N-doped ZnO and N-doped ZnO/CNT thin film photocatalysts.

**Figure 5 molecules-26-01470-f005:**
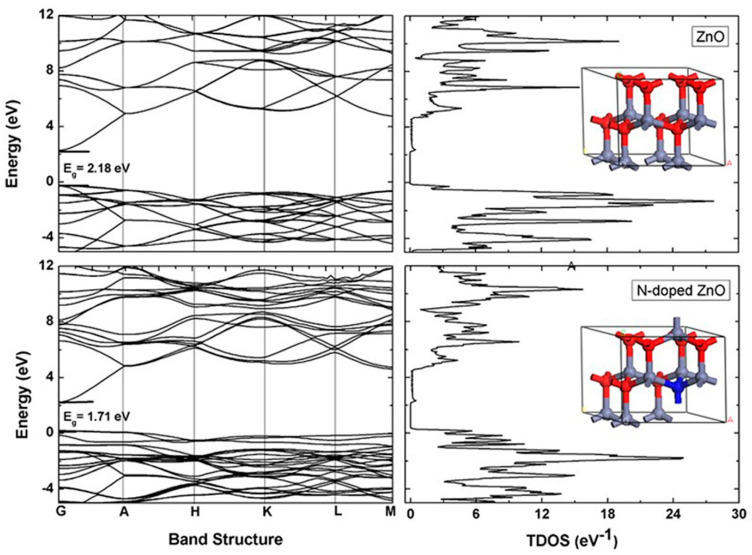
Calculated electronic band structure and total density of states of ZnO (**top**) and N-doped ZnO (**bottom**) using the DFT-GGA+U method. The inset figure shows the optimized structure of ZnO (**top**) and N-doped ZnO (**bottom**) with periodic boundary condition. Red, blue yonder, and blue represent oxygen, zinc and nitrogen atom, respectively.

**Figure 6 molecules-26-01470-f006:**
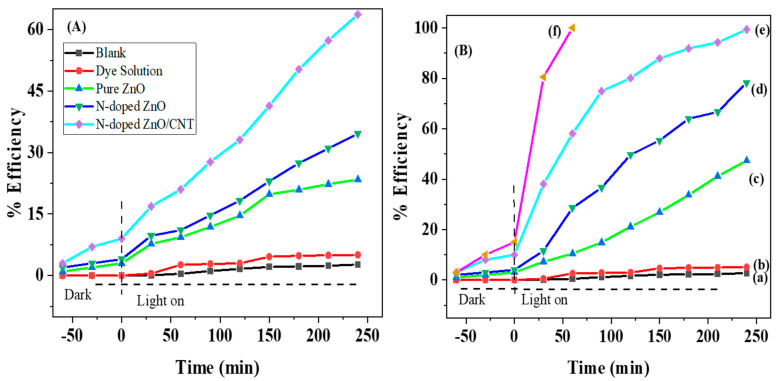
(**A**) Photo-catalytic degradation efficiencies of methylene blue under Visible light irradiation, (**B**) under UV light irradiation: (**a**) blank; (**b**) dye solution; (**c**) pure ZnO; (**d**) N-doped ZnO; (**e**) N-doped ZnO/CNT; and (**f**) N-doped ZnO/CNT + H_2_O_2_, respectively.

**Figure 7 molecules-26-01470-f007:**
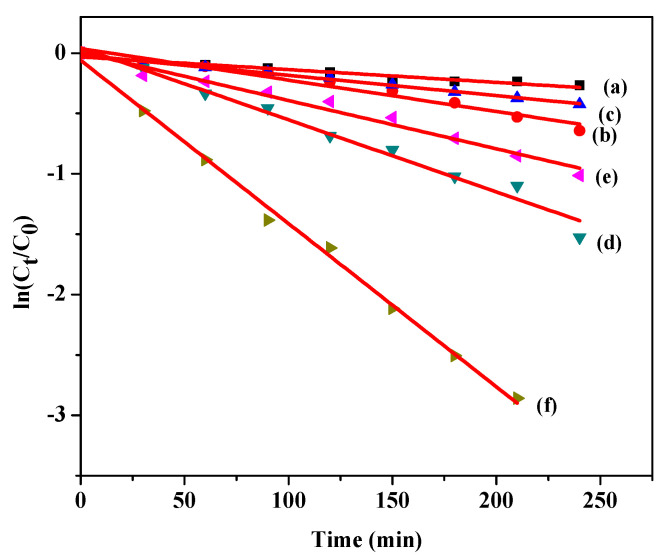
First order kinetics of MB dye degradation by Pure ZnO (**a**) visible light & (**b**) UV light; N-doped ZnO (**c**) visible light & (**d**) UV light; and N-doped ZnO/CNT (**e**) visible light & (**f**) UV light irradiation, respectively.

**Figure 8 molecules-26-01470-f008:**
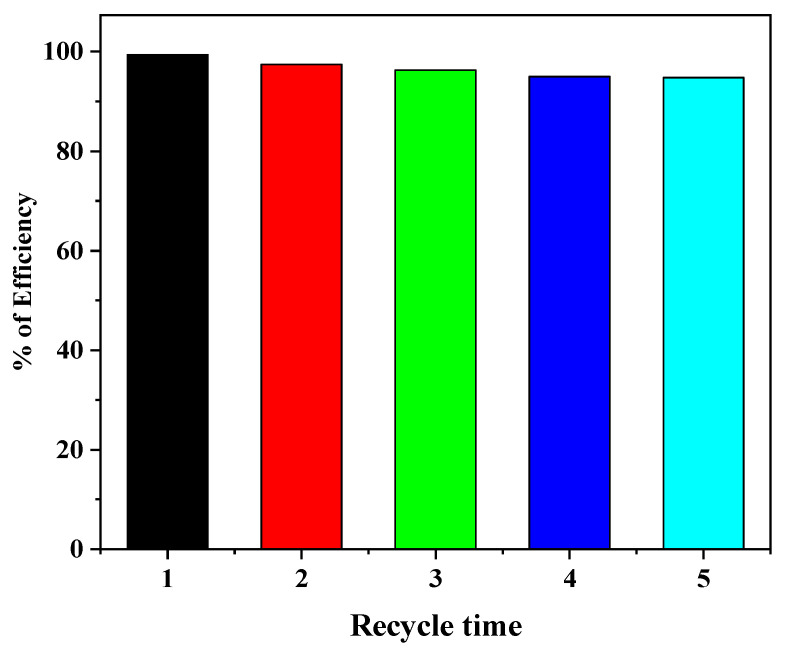
Recycle and reuse of photocatalyst for MB degradation under UV light irradiation (MB concentration: 5 ppm; photocatalyst: N-doped ZnO/CNT; solution pH 6.4; irradiation time: 4 h).

**Figure 9 molecules-26-01470-f009:**
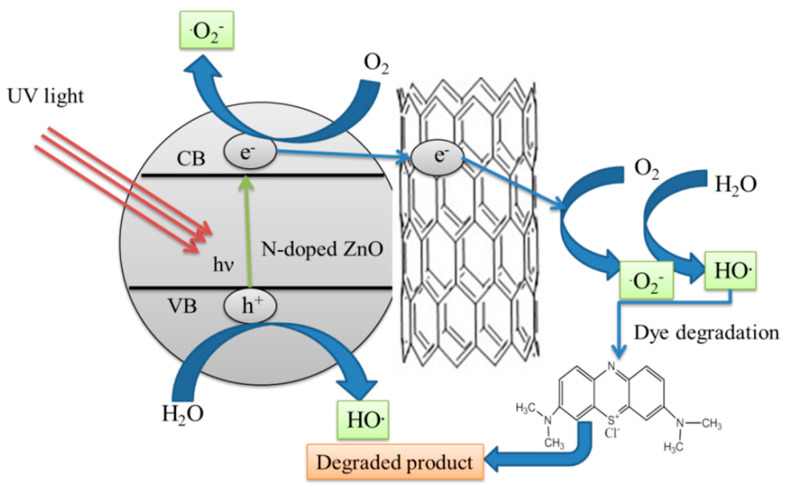
Schematic illustration of photocatalytic mechanism of the N-doped ZnO/CNT thin films under UV light irradiation.

**Figure 10 molecules-26-01470-f010:**
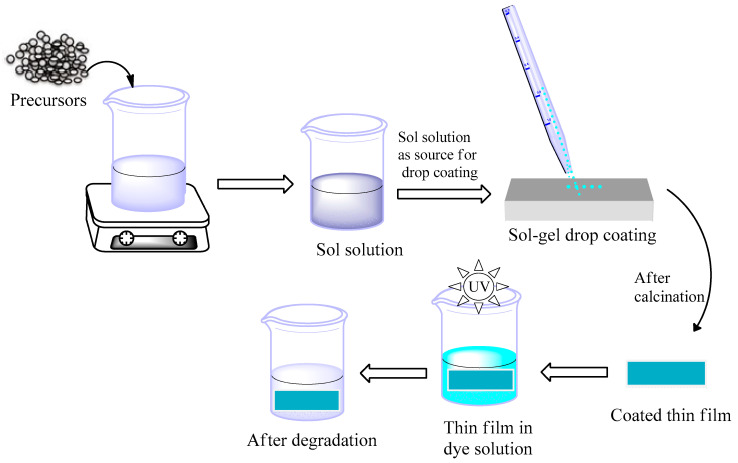
Schematic diagram for the fabrication of N-doped ZnO/CNT thin film on glass substrate by sol-gel drop coating process.

**Table 1 molecules-26-01470-t001:** Representation of lattice size, lattice strain and dislocation density of pure ZnO, N-doped ZnO and N-doped ZnO/CNT thin film.

Sample	Lattice Size (D)(nm)	Lattice Strain [(ε) × 10^−4^]	Dislocation Density (δ) (lines/m^2^)
Pure ZnO	29.07	30.71	1.18 × 10^15^
N-doped ZnO	27.13	33.73	1.36 × 10^15^
N-doped ZnO/CNT	14.28	70.64	4.90 × 10^15^

**Table 2 molecules-26-01470-t002:** Lattice parameters of Pure ZnO, N-doped ZnO, and N-doped ZnO/CNT thin film.

Sample	Lattice Constant a = b (Ǻ)		Lattice Constant c (Ǻ)		c/a Ratio	Unit Cell Volume (V) (Ǻ^3^)	
	Calculated	ASTM	Calculated	ASTM		Calculated	ASTM
Pure ZnO	3.2340	3.250	5.1842	5.207	1.6030	46.9579	47.621
N-ZnO	3.2374		5.1956		1.6048	47.1599	
N-ZnO/CNT	3.2467		5.1899		1.5985	47.3786	

**Table 3 molecules-26-01470-t003:** Values of rate constant (k) and correlation coefficient (r^2^) for MB dye degradation kinetics.

Thin Film Photocatalyst	Rate Constant, k(min^−1^)	Correlation Coefficients (r^2^)
(a) Bare ZnO under vis	0.0010	0.94376
(b) Bare ZnO under UV	0.0026	0.96763
(c) N-doped ZnO under vis	0.0017	0.98564
(d) N-doped ZnO under UV	0.0060	0.97787
(e) N-doped ZnO/CNT under vis	0.0040	0.96608
(f) N-doped ZnO/CNT under UV	0.0135	0.99619

**Table 4 molecules-26-01470-t004:** Comparison of degradation percentage of N-doped ZnO/CNT composite photocatalysts with previously reported composite photocatalysts.

Type of Photocatalyst	Model Pollutant	Conc.	Light Source	Illumination Time min	^a^ DE in %	Ref. Photocatalyst; ^a^ DE in %	Reusability	Ref.
ZnO-rGO	MB	5 mg	500 W Xe	260	88	Pure ZnO; 68	-	[17]
ZnO-CNT	MO	10 mg	60 W Hg	260	98	Pure ZnO, 35	4 cycles	[53]
g-C_3_N_4_/ZnO	RhB	0.05 g	500 W Xe	100	97.4	Pure ZnO; 48	5 cycles	[54]
Mg-ZnO/CNT	MB	10 mg	150 W Xe	60	84	Pure ZnO; 20	5 cycles	[55]
Cd-ZnO/CNT	MO	20 mg	15 W Hg	110	93	ZnO/CNT; 44	-	[39]
N-ZnO/C-dots	MG	20 mg	150 W Vis	160	85	Pure ZnO; 65	-	[56]
N-ZnO/CNT	MB	5 mg	60 W Hg	240	100	Pure ZnO, 47.36	5 cycles	This work
N-ZnO/CNT	MB	5 mg	200 W Tungsten	240	70	Pure ZnO, 25		This work

^a^—degradation efficiency.

## Data Availability

Not applicable.

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
