# Peer review of "A Highly Efficient and Stable Photocatalyst; N-Doped ZnO/CNT Composite Thin Film Synthesized via Simple Sol-Gel Drop Coating Method"

_molecules, 2021, doi:10.3390/molecules26051470_

Round 1

Reviewer 1 Report

The manuscript on urea-modified ZnO/CNT has some interesting elements, but has not been written well, and thus should be carefully improved before possible acceptance, as shown below:

1)

Results should be better discussed. For example, although authors have calculated lattice parameters, the shift in XRD peak could confirm the doping of ZnO with nitrogen. I think that main peak of ZnO could be compared in these three samples. Moreover, modification with CNT has resulted in a decrease in crystalline properties of the material, and this has not been discussed at all.

2)

Authors should check other reports on similar photocatalysts, e.g., by Mitoraj and Kisch. Other possibilities should be discussed, e.g., melem formation (from urea). It is well known that also in the case of titania modification with urea, at first doping was considered, but then formation of melem-like species have been confirmed.

3)

I cannot see any CNT in SEM images. Additionally, all SEM images look same – just different magnification. I think it is necessary to shown nanotube structure.

4)

Photoabsorption data look like UV/vis spectra rather than diffuse-reflectance spectroscopy data. I think it is impossible to estimate bandgap energy based on these spectra.

5)

I am wonder why there is any vis activity of bare ZnO photocatalyst. Authors should know that testing of vis activity for dyes is not recommended due to dye-sensitization mechanisms (as proven for MB e.g., by Ohtani). Moreover, I am not sure about emission of tungsten lamp – some tungsten lamps imit from 350 nm.

6)

The differences in dark adsorption should be also shown and discussed.

7)

Some sentences and phrases are difficult for understanding, e.g.,

i) “The hexagonal crystalline structure was confirmed from XRD measurement without any other impurity phase detection/present/presence in samples.” – It is unclear for which component the hexagonal structure has been confirmed. Moreover, “phase detection/precent/presence” sounds very strange. Similar sentence structure has also been used in other parts, and I think it is unclear why authors want to write in this way, e.g., “activity/behavior/ performance”. I think either “activity” or “behavior” or “performance” can be used.

ii) I think that there is no need to use the world “form” in the title.

iii) There is no need to capitalize “Photo” (e.g., line 166).

Author Response

1st Reviewer

The manuscript on urea-modified ZnO/CNT has some interesting elements, but has not been written well, and thus should be carefully improved before possible acceptance, as shown below:

Sincere gratitude to the reviewer for recommending this work for possible publication. Herein, we have tried our best to explain all the queries which are raised by respected reviewer and subsequently changed the manuscript as per recommendation.

  1. Comment: Results should be better discussed. For example, although authors have calculated lattice parameters, the shift in XRD peak could confirm the doping of ZnO with nitrogen. I think that main peak of ZnO could be compared in these three samples. Moreover, modification with CNT has resulted in a decrease in crystalline properties of the material, and this has not been discussed at all.

Reply: We would like to thank the reviewer for his scholastic suggestion. According to your suggestion we’ve revised the XRD findings with details explanation. The details explanation, particle size reduction due to the addition of CNT, has been discussed in the revised manuscript, marked with red color.

  1. Comment: Authors should check other reports on similar photocatalysts, e.g., by Mitoraj and Kisch. Other possibilities should be discussed, e.g., melem formation (from urea). It is well known that also in the case of titania modification with urea, at first doping was considered, but then formation of melem-like species have been confirmed.

Reply: Being grateful to the reviewer’s judicial comment, we would like to inform you that we have synthesized pure ZnO, N-doped ZnO and N-doped ZnO/CNT nanocomposite thin film according to the reference no-42 with some modification. Here the calcination temperature was 500 °C and from FTIR results we didn’t get any characteristics peak for melem formation from urea. A detail experimental study is necessary to make exclusive conclusion about the formation of melem species during the modification of ZnO by urea at various calcination temperature like 300 - 600 °C. However, within a short period and the world crucial situation for Covid-19, we are unable to extend the experimental work for this time. We will definitely perform new experiments considering reviewer judicial suggestion in near future for our next report.

  1. Comment: I cannot see any CNT in SEM images. Additionally, all SEM images look same – just different magnification. I think it is necessary to shown nanotube structure.

Reply: Being respectful to your suggestion, we’ve added the new FESEM image of N-doped ZnO/CNT composite thin film in the revised manuscript (please see Fig.3 (c). We have revised the manuscript accordingly.  Although it is not very clear because we used very low amount of CNT (5wt%) in the fabrication process.

  1. Comment: Photoabsorption data look like UV/vis spectra rather than diffuse-reflectance spectroscopy data. I think it is impossible to estimate band gap energy based on these spectra.

Reply: Thanks to the reviewer for this valuable suggestion. We are fully agreed with the reviewer. However, in this manuscript we have calculated the material’s band gap using the well-known tauc equation from UV-Vis absorption spectra. Then we compared the values by calculating the band gap with the help of well-known equation (Eg= 1240 / λ). We got the similar results in both cases. However, we do not have the facility to take the diffuse reflectance spectra of the materials. For this reason, we could not be able to include the reflectance data to determine the band gap in our manuscript. The authors greatly appreciate and have noted the point raised by reviewer and hope that it will be included in our future article.

  1. Comment: I am wonder why there is any vis activity of bare ZnO photocatalyst. Authors should know that testing of vis activity for dyes is not recommended due to dye-sensitization mechanisms (as proven for MB e.g., by Ohtani). Moreover, I am not sure about emission of tungsten lamp – some tungsten lamps imit from 350 nm.

Reply: Being grateful to the reviewer’s judicial comment, we would like to inform that first we checked absorption ability using only glass slides, i.e. blank experiment (without catalyst) and then we did the same experiment using dye under visible light irradiation (please see figure 7). Interestingly we couldn’t get any appreciable change due to adsorption as well as photolysis. Many researchers have been reported on bare ZnO visible light photoactivity of MB dye [25]. In our experiment we have used the visible light that has the Intensity of 2150 lm with a wavelength range of 500–620 nm.

  1. Comment: The differences in dark adsorption should be also shown and discussed.

Reply: Many thanks to the reviewer for this point. The differences in dark adsorption have been shown and discussed (please see Figure 7).

  1. Comment: Some sentences and phrases are difficult for understanding, e.g.,
  2. i) “The hexagonal crystalline structure was confirmed from XRD measurement without any other impurity phase detection/present/presence in samples.” – It is unclear for which component the hexagonal structure has been confirmed. Moreover, “phase detection/precent/presence” sounds very strange. Similar sentence structure has also been used in other parts, and I think it is unclear why authors want to write in this way, e.g., “activity/behavior/ performance”. I think either “activity” or “behavior” or “performance” can be used.
  3. ii) I think that there is no need to use the world “form” in the title.

iii) There is no need to capitalize “Photo” (e.g., line 166).

Reply: According to your suggestion, we’ve corrected the sentences in the revised manuscript, marked with red color. We have also removed the word “form” from the title in the revised manuscript.

Reviewer 2 Report

The authors fabricated the thin film of N-doped ZnO/CNT nanocomposite by a low cost sol-gel drop coating protocol and used to achieve photo-catalytic degradation of MB dye. However, there are some problems need to be addressed. I suggest that the article be reconsidered after major revision:

  1. The abstract section is too cumbersome and the authors need to refine the content of the abstract section.
  2. In your introduction, it is not reasonable to introduce the degradation of MB as the photocatalytic effect alone, and it is not too innovative. The solution of water pollution is the key point of photocatalytic oxidation technology. You can refer to “Journal of Materials Science & Technology, 2020, 56, 45-68”, “Applied Surface Science, 2019, 463, 556–565” and “Chemical Engineering Journal, 2021, 406, 126844”
  3. The trace of CNT cannot be visually seen in the FESEM figure. Please provide the valid figures again.
  4. The DES data is not reliable, especially for the tangent part to determine the bandgap values, please retest.
  5. Please draw the data of visible light degradation and ultraviolet light degradation separately, it looks too messy.
  6. The degradation of a single colored dye does not fully demonstrate the feasibility of the photocatalysis, or it may be due to the photosensitization of MB resulting in catalytic activity. The author should degrade some colorless organic matter or antibiotics to enhance the argument. You can refer to “Chemical Engineering Journal, 2021, 411, 128615.”, “Applied Surface Science, 2018, 427, 1046-1053” and “Water 2021, 13, 288”.

Author Response

The authors fabricated the thin film of N-doped ZnO/CNT nanocomposite by a low cost sol-gel drop coating protocol and used to achieve photo-catalytic degradation of MB dye. However, there are some problems need to be addressed. I suggest that the article be reconsidered after major revision:

Sincere gratitude to the reviewer for the suggestions to improve the quality of our manuscript.

  1. Comment: The abstract section is too cumbersome and the authors need to refine the content of the abstract section.

Reply: Thanks to the reviewer for your valuable suggestion. According to your suggestion we’ve refined the content of the abstract part in the revised manuscript.

  1. Comment: In your introduction, it is not reasonable to introduce the degradation of MB as the photocatalytic effect alone, and it is not too innovative. The solution of water pollution is the key point of photocatalytic oxidation technology. You can refer to “Journal of Materials Science & Technology, 2020, 56, 45-68”, “Applied Surface Science, 2019, 463, 556–565” and “Chemical Engineering Journal, 2021, 406, 126844”

Reply: We are fully agreed with the reviewer’s judicial comment. We’ve added some explanation based on suggested references [1-3] in the introduction section, marked with red color (please see page 2 line no. 30-39 and 43-46,  page 4 line no. 92-93). We have included those references in our manuscript.

  1. Comment: The trace of CNT cannot be visually seen in the FESEM figure. Please provide the valid figures again.

Reply: Being respectful to your suggestion, we’ve added the new FESEM image of N-doped ZnO/CNT composite thin film in the revised manuscript (please see Fig.3 (c). Although it is not very clear because we used very low amount of CNT (5wt%) in the fabrication process.

  1. Comment: The DES data is not reliable, especially for the tangent part to determine the bandgap values, please retest.

Reply: Thanks to reviewer for this valuable suggestion. In this manuscript we have calculated the material’s band gap using the well-known tauc equation from UV-Vis absorption spectra. Then we compared the values by calculating the values with the help of equation (Eg= 1240 / λ). We got the similar results in both cases. However, we do not have the facility to take the diffuse reflectance spectra of the materials.  For this reason, we could not be able to include the reflectance data to determine the band gap in our manuscript. The authors greatly appreciate and have noted the point raised by reviewer and hope that it will be included in our future research article.

  1. Comment: Please draw the data of visible light degradation and ultraviolet light degradation separately, it looks too messy.

Reply: As per your wise suggestions, we have drawn the data of visible light degradation (Fig. 7(a)) and ultraviolet degradation (Fig. 7(b)) separately (Please see the Fig.7).

  1. Comment: The degradation of a single colored dye does not fully demonstrate the feasibility of the photocatalysis, or it may be due to the photosensitization of MB resulting in catalytic activity. The author should degrade some colorless organic matter or antibiotics to enhance the argument. You can refer to “Chemical Engineering Journal, 2021, 411, 128615.”, “Applied Surface Science, 2018, 427, 1046-1053” and “Water 2021, 13, 288”.

Reply: Being respectful to your judicial query, for this purpose we have carried out the blank sample as well photolysis reaction of MB dye and we couldn’t observe any significant change due to adsorption and photolysis. Moreover, it would be great if we do some degradation experiments by considering colorless organic matter or antibiotics. Unfortunately it’s quite impossible to do these degradation experiments within a short period of time during this covid-19 pandemic situation. Although we have added some explanation about degradation based on your suggested references [2, 53, 54], marked with red color. We have included those references in our manuscript. The authors also appreciate and have noted the point raised by the reviewer, and hope that the extensive explanation will be a subject of future article concerning degradation of some organic colorless matter or antibiotics using bare ZnO, N-doped ZnO and N-doped ZnO/CNT.

Round 2

Reviewer 1 Report

I am sorry, but I cannot see nanotubes' structure at all. maybe authors should change the title and descriptions since it is practically impossible to see CNT.

Author Response

We are very grateful to the reviewers for their scholastic and valuable comments on our manuscript.

Response to the reviewer’s comments:

Reviewer 1 

Comments and Suggestions for Authors

I am sorry, but I cannot see nanotubes' structure at all. maybe authors should change the title and descriptions since it is practically impossible to see CNT.

Response: We are very grateful to the reviewer for the positive comments and constructive recommendation to improve our manuscript. Considering his scholastic recommendation, we have revised our manuscript by including the following sentence in the section 3.2:

“However, it looks very difficult to figure out the CNT structure from Fig. 3c with that low resolution image”

We will definitely provide high resolution image of FESEM with EDX profile for the clear detection of CNT in our next submission.

Reviewer 2 Report

The authors have revised the manuscript as required and suggest acceptance.

Author Response

Review 2

Comments and Suggestions for Authors

The authors have revised the manuscript as required and suggest acceptance.

Response: We are very grateful to the reviewer to accept our manuscript for publication.